# Bisindolyl Maleimides and Indolylmaleimide Derivatives—A Review of Their Synthesis and Bioactivity

**DOI:** 10.3390/ph16091191

**Published:** 2023-08-22

**Authors:** Louise N. Cooney, Kevin D. O’Shea, Hannah J. Winfield, Michael M. Cahill, Larry T. Pierce, Florence O. McCarthy

**Affiliations:** School of Chemistry and ABCRF, University College Cork, Western Road, T12K8AF Cork, Ireland; 113352196@umail.ucc.ie (L.N.C.); kevin_oshea@umail.ucc.ie (K.D.O.); 106432930@umail.ucc.ie (H.J.W.); mickcahill72@hotmail.com (M.M.C.); lpierce9184@yahoo.co.uk (L.T.P.)

**Keywords:** indole, bisindolyl, maleimide, BIM, fragment-based design, kinase inhibition

## Abstract

The evolution of bisindolyl maleimides and indolyl maleimide derivatives and their unique biological activities have stimulated great interest in medicinal chemistry programs. Bisindolylmaleimide (BIM)-type compounds arise from natural sources such as arcyriarubin and are biosynthetically related to indolocarbazoles. BIMs are commonly the immediate synthetic precursors of indolocarbazoles, lacking a central bond between the two aromatic units and making them more flexible and drug-like. Synthetic endeavours within this class of compounds are broad and have led to the development of both remarkably potent and selective protein kinase inhibitors. Clinical BIM examples include ruboxistaurin and enzastaurin, which are highly active inhibitors of protein kinase C-β. While BIMs are widely recognised as protein kinase inhibitors, other modes of activity have been reported, including the inhibition of calcium signalling and antimicrobial activity. Critically, structural differences can be used to exploit new bioactivity and therefore it is imperative to discover new chemical entities to address new targets. BIMs can be highly functionalised or chemically manipulated, which provides the opportunity to generate new derivatives with unique biological profiles. This review will collate new synthetic approaches to BIM-type compounds and their associated bioactivities with a focus on clinical applications.

## 1. Introduction

Bisindolylmaleimides (BIMs) are widely recognised for their activity against protein kinases and from a synthetic perspective can be highly functionalised or chemically manipulated. This provides the opportunity to generate novel analogues and derivatives with unique biological profiles. Although BIMs show significant activity themselves and serve as targets in their own right, they are also important precursors in the synthesis of the indolocarbazole compound class.

Over the years, bisindolylmaleimides have been identified as reference compounds to benchmark a number of bioassays, including kinase inhibition, and a summary of their structure and diversity is shown in Figure 1 [1]. It is evident that the maleimide functional group is key to their activity, with only one (**BIM-V**) *N*-alkylated at this position. It is also clear that alkylation of the indole nitrogens (one or both) contributes important characteristics to BIM activity. Surprisingly, while a number of reviews for the related indolocarbazole class of compounds exist, no focused review of the BIM class has been undertaken [2,3,4,5,6,7,8]. In setting the scope for this review, bisindolyl, benzofuranylindolyl and naphthylindolyl maleimides are incorporated but extended heterocycles such as azaindolyl, benzisoxazolyl, imidazopyridinyl and pyrazolopyridinyl indolyl maleimides have been omitted as this field is deserving of a review in itself. In addition, the biosynthesis of BIMs has already been covered elsewhere [2].

This review sets out to describe the synthetic routes commonly used to generate BIMs and their derivatives and to catalogue the relevant bioactivity of these multifunctional molecules. In Section 2, we will summarise the methods used to make diverse BIMs, and in Section 3, the recorded biological activity of diverse BIMs uncovering kinase and non-kinase effects with a focus on their solved crystal structures and clinical applications.

## 2. Synthesis of Bisindolylmaleimides and Derivatives

The synthesis of BIMs can be classified into those methods which substitute an existing maleimide ring and those that form the maleimide ring in the final steps. Once formed, BIMs can be modified to yield highly functionalised compounds and those that have clinical relevance.

### 2.1. BIMs via the Use of Preformed Maleimides

These routes typically involve indole electrophilic substitution where the indoles are initially treated with a Grignard reagent or non-nucleophilic base to activate the C-3 position, prior to the reaction with 2,3-dihalomaleimides. The synthesis of bisindolylmaleimides was first described by Steglich et al. in 1980 (Figure 1) [9] in an investigation of the fruiting bodies of the slime mould, *Arcyria denudate*. Utilising Grignard methodology, the *N*-methyl maleimide **3** was formed through the reaction of *N*-methyl-2,3-dibromomaleimide **1** and indolyl magnesium iodide **2** in 60% yield in the presence of hexamethylphosphoramide (HMPA).

This method was later applied by Weinreb and Kaneko for their syntheses of benzyl-protected staurosporinone and rebeccamycin, respectively [10,11]. Weinreb et al. coupled indole magnesium bromide with *N*-benzyl-2,3-dibromomaleimide and HMPA in THF to form the BIM intermediate in 54% yield. Kaneko et al. coupled 7-chloroindole with *N*-benzyloxymethyl-2,3-dibromomaleimide in benzene containing a catalytic amount of HMPA and the desired BIM intermediate was formed in 27% yield.

Brenner et al. achieved the synthesis of unsymmetrically substituted arcyriarubin B **9** for the first time by incorporating a maleimide NH consistent with the most bioactive BIMs (Figure 2) [12]. They found that their initial reaction of indole Grignard **4** with 2,3-dibromo-*N*-methylmaleimide **1** is strongly solvent-dependent. For example, in ether/benzene mixtures, this gives a mixture of the mono- and bisindolylmaleimides, toluene gives the bisindolyl compound **3** in 70% yield and THF gives the monoindolyl compound **5** in 74% yield. This finding allowed for the preparation of an unsymmetrically substituted bisindolylmaleimide through Boc protection followed by a reaction with 2 equivalents of substituted indole Grignard **6** in THF to give the Boc-protected **7** in 85% yield. Deprotection was carried out by heating the compound to 180 °C to give **8**. Conversion of the methylmaleimide to the maleimide was then carried out using forcing conditions to give arcyriarubin B **9**.

Harris et al. attempted the use of dibromomaleimide **10** directly in the Grignard reaction to access the structure of arcyriarubin A (**BIM-IV**); however, a much lower yield of 29% was obtained, which further consolidated the use of *N*-alkylated maleimides in this step (Figure 3) [13].

The Grignard step was used in a route towards indolocarbazole staurosporine (**14**) by Xie and Lown in 1994 (Figure 4). In this case, an *N*-benzyl-protected 2,3-dibromomaleimide **11** was employed in a THF/toluene mixture to generate the bisindolylmaleimide **12** in 64% yield and was converted to its corresponding maleic anhydride **13** in high yield [14]. Multiple groups have subsequently utilised the anhydride as a BIM intermediate (see below).

Faul et al. investigated the Grignard route in significant detail [15]. They found that dichloromaleimides **15**, which are more economical and readily available, have comparable reactivity to the dibromo analogues. They also found the synthesis to be solvent-dependent and identified the optimal solvent system toluene/ether/THF (5:1:1) for this reaction. The addition of aqueous ammonium chloride to the reaction mixture eliminated the need for chromatography and, on the incorporation of all these improvements, the synthesis of the *N*-methylmaleimide **3** was impressively scaled up to 770 g with a yield of 72% (Figure 5).

The necessity for a large excess of the indole Grignard reagent **4** to obtain a high yield (due to consumption of the indolylmagnesium bromide) led Ohkubo et al. to examine the use of indoles with several bases, including *n*-BuLi, LDA, LiHMDS, NaHMDS and KHMDS, as an alternative, finding LiHMDS to be the most effective [16]. They found that just one equivalent of indole is sufficient for indolylation when using a non-nucleophilic base and used this method in the synthesis of arcyriaflavin B, arcyriaflavin C and arcyriaflavin D (Figure 6). One equivalent of 6-benzyloxyindole **16** and **1** were coupled using two equivalents of LiHMDS in THF to give the monoindolyl **17** in 93% yield. On base screening with one equivalent of indole, the use of LDA also gave a relatively high yield of 76%, but NaHMDS and KHMDS gave lower yields with *n*-BuLi only giving a trace of the desired product. Protection with a Boc group and reaction with several different indoles in the presence of two equivalents of LiHMDS again formed the corresponding bisindole compounds **19** in good yields. These were subsequently converted to Arcyriaflavins B–D.

Neel et al. employed a combination between maleimide forming and maleimide substitution to use a Suzuki cross-coupling reaction as a key step from an indolylmaleimide triflate intermediate **23** (Figure 7) [17]. This intermediate was obtained via the reaction of amide **20** with dimethyl oxalate **21** in the presence of potassium *tert*-butoxide in DMF to form maleimide **22**, which was subsequently methylated and converted to the triflate **23**. The triflate was then coupled with protected 3-indole boronic acid **24** in a Suzuki reaction to give the desired bisindolylmaleimide **25** in 55% yield. The tosyl group was then removed yielding the unsymmetrical bisindolylmaleimide **26**.

In 2010, Wang et al. also designed a three-step route towards arcyriaflavin A using a Suzuki cross-coupling method with indole nitrogen protection [18]. This required the preparation of the indole boronic acid **24** which was reacted with the *N*-methyl dibromomaleimide **1** to generate the *N,N′*-bistosylated maleimide **27** (Figure 8). Following this, subsequent cleavage of the *N*-protecting groups at reflux in basic solution afforded the maleic anhydride **13** in 82% yield. Treatment of the maleic anhydride with HMDS furnished the bisindolylmaleimide arcyriarubin A (**BIM-IV**).

In 2017, Shengyin et al. developed a cost-effective process to obtain arcyriarubin A (**BIM-IV**) in high yield and purity (Figure 9) [19]. Indole **28** was reacted with maleimide **29** in the presence of zinc chloride to obtain the succinimide intermediate **30**. Treatment with DDQ yielded the mono-indole maleimide **31** in excellent yield. After extensive scoping of the reaction, it was found that a high yield of 77% for **BIM-IV** could be obtained by utilising palladium acetate as the catalyst, copper acetate as the oxidant and two molar equivalents of indole. The authors report that this is efficient, economical, has a short reaction time and utilises simple techniques while retaining high yield and purity.

### 2.2. BIMs via the Formation of the Maleimide Ring

Though much research has focused on maleimide substitution as a route to BIMs, another highly reported approach is through maleimide ring formation.

In 1990, Davis et al. reported a new route to indolylaryl maleic anhydrides using a Perkin condensation between indolyl-3-glyoxylyl chlorides and aryl acetic acids [20]. For the synthesis of bisindolyl compound **35**, methyl indole **32** was treated with oxalyl chloride to form 2-(1-methylindol-3-yl)-2-oxoacetyl chloride **33** which was then used directly in the reaction with 1-methyl-3-indolylacetic acid **34** in the presence of triethylamine to give the maleic anhydride in 36% isolated yield (Figure 10). One disadvantage of this route is that the reaction requires the protection of the indole nitrogens, but its discovery has enabled the facile design and preparation of indole *N*-functionalised bisindolylmaleimides.

Later that year, the group reported a mild conversion of maleic anhydrides **35** into maleimides **36** using HMDS/MeOH (Figure 11) [21]. This procedure gave excellent yields for a variety of maleic anhydrides and did not affect more sensitive functional groups such as nitriles and esters. This reaction involved the hydrolysis of the hexamethyldisilazane (HMDS) reagent with methanol, which allows the release of ammonia in situ, and afforded the bisindolylmaleimide **36** in excellent yield.

This further improved the route from alkylated maleimides (e.g., **37**) which were converted to NH maleimides through maleic anhydride. This had previously required the addition of ammonia, or an ammonia source, whilst heating the reaction mixture to a temperature of 140 °C [22]. These harsh conditions were not suitable for maleimides containing sensitive functionality, and even *N*-alkylated anhydrides produced a mixture of products containing little or none of the desired maleimide.

In 1993, Bit et al. developed a new coupling reaction which allowed direct access to the bisindolylmaleimide system involving the reaction of indole oxoacetyl chlorides with acetimidates (Figure 12) [23]. The reaction of **33** with an indole-3-carboximidate **40** in the presence of excess triethylamine gave the hydroxypyrrolone **41**, which underwent dehydration and hydrolysis to the maleimide **36** upon treatment with *p*-toluenesulfonic acid, and a panel of *N*-substituted bisindolylmaleimides was synthesised. This route tolerates an unprotected indole nitrogen on the imidate and has a higher yield than the Perkin route based on the indole starting material but suffers from low acetimidate availability.

In 1998, Faul et al. described a synthesis of **BIM-IV** which involved the assembly of the maleimide ring (Figure 13) [24]. Indole **28** was treated with oxalyl chloride to generate indole-3-oxoacetyl chloride in situ, and then treated with sodium methoxide to afford **42**. Oxoacetate ester **42** was then combined with acetamide **43** in a Perkin-type condensation in the presence of potassium *tert*-butoxide in THF. This methodology exhibited a significant increase in the overall yield in comparison to other methodologies while using relatively stable precursors. As indolyl-3-oxoacetyl chlorides were successful in Davis’s Perkin condensation, the maleimide formation was attempted directly with the oxoacetyl chloride instead of the ester but the yield dropped from quantitative to 68%.

This reaction was found to tolerate an unprotected indole nitrogen on either starting material and with alkyl *N*-substituents containing a variety of functional groups (OH, OTr and NMe_2_) with conversions ranging from 84% to 100%. The main disadvantage of the method is that indole-3-acetamides are relatively expensive.

In 2000, Bergman and Pelcman developed a biomimetic synthesis via oxidative coupling of the dianion derived from 2-(indol-3-yl)methylacetate **44** (Figure 14) [25]. Diester **45** was formed by the addition of 2 equivalents of LDA to **44** in THF, followed by coupling with 0.5 equivalents of iodine and an acidic workup. Heating with benzylamine gave succinimide, which upon oxidation with DDQ afforded the maleimide **12**.

In 2006, Roy et al. developed a method for the synthesis of bisindolylmaleimide **36**, along with other indolylaryl and indolylheteroaryl maleimides, by reacting *N*-methylindole-3-glyoxylamide **46** with methyl aryl acetates (e.g., **47**) in the presence of potassium *tert*-butoxide in THF (Figure 15) [26]. Oxoacetamide **46** was prepared from *N*-methylindole **32** through treatment with oxalyl chloride followed by aqueous ammonium hydroxide. This method was found to be successful for a range of aryl and heteroaryl derivatives, although unsuitable for an unprotected indole.

As can be seen in Figure 1, alkylation of the indole nitrogen is a key aspect of active BIMs, and the formation of mono-, bis- and di-alkylated BIMs is of direct biological and clinical importance. Synthetic routes have been developed which incorporate indole alkylation in the starting materials or where the alkylation may be added at a late stage in the presence of a maleimide/maleic anhydride ring. An example of this is seen in the 2015 study by Gao et al., who generated BIM-I (GF109203X) using a methodology similar to Figure 13 without the use of protecting groups [27]. More recently, Li et al. and Winfield et al. simultaneously demonstrated that the maleic anhydride intermediate **51** could be formed directly via a Perkin-type condensation method and modified to selectively generate mono-, bis- and di-alkylated derivatives **53** (Figure 16) [28,29]. Indole oxoacetate salts **49** and acetic acid components **50** were combined to furnish the maleic anhydride headgroup which was used to develop maleimides and further BIM derivatives. This route allows for full functionality of the BIM framework and late-stage diversification.

### 2.3. Synthesis of Clinically Relevant BIM Kinase Inhibitors

Enzastaurin (**57**) is a prime example of a clinically used bisalkylated, unsymmetrical bisindolylmaleimide. Both indole precursors are functionalised prior to the maleimide-forming coupling step. The synthesis of **57** was detailed by Faul et al. in 2003 (Figure 17) [30]. In the initial step, the indole precursor **54** was converted to its hydrochloride salt before the addition of oxalyl chloride and methanol, which furnished the oxoacetate ester **55**. This ester moiety was then condensed with 2-(1-methyl-1*H*-indol-3-yl)acetamide (**56**) in a Perkin-type condensation, as seen in Figure 13, to furnish enzastaurin (**57**).

A separate synthetic strategy was also investigated, whereby the indole *N*-functionalities of **55** and **56** were exchanged. This involved the coupling of the 2-(1-methyl-1*H*-indol-3-yl)-2-oxoacetate component with the bulky substituted indole-3-acetamide, under the same condensation conditions. However, in this case, a lower yield of enzastaurin (**5*7***) (53%) was obtained.

The macrocyclic BIM ruboxistaurin (**61**) (LY317644) is a selective PKC inhibitor and its synthesis was reported by Jirousek et al. in 1995 (Figure 18) [31]. The bisindolylmaleimide **3** was generated by Steiglich’s Grignard method (Figure 1) and used as the precursor for this synthetic route. The alkylating agent **58** was prepared from dimethyl 2-hydroxysuccinate for the initial synthetic step in the sequence. A combination of **3** and **58** in base conditions furnished the macrocycle **59** with an excellent yield.

A base hydrolysis step enabled the cleavage of the silyl-protecting group as well as the conversion to the corresponding maleic anhydride. The maleic anhydride intermediate was not isolated but directly transformed to its free *N*-H maleimide **60** [31]. This penultimate intermediate was employed in the final step of this sequence, which involved the mesylation of the alcohol group followed by displacement with dimethylamine to afford the macrocyclic ruboxistaurin (**61**) in a 67% yield [32].

### 2.4. Preparation of Selected Modified Indolylmaleimides

Bisindolylmaleimides are an attractive pharmacophore and they have been widely explored in the literature since their discovery in the early 1980s [1]. Therefore, as seen below with 7-azaindole, it is of interest to include non-indole components in this scaffold to probe if aryl replacements enhance biological effects.

As an example of the use of 7-azaindole to form BIMs, in 2003, Kuo et al. investigated the structural template of the PKC selective isoform inhibitor ruboxistaurin (**61**) and designed a panel of macrocyclic derivatives to probe kinase inhibition [33]. One or both of the indole components was replaced with 7-azaindole and resulted in the discovery of a bisazaindole compound (**66**) which had surprisingly selective activity against GSK-3β kinase, rather than the anticipated PKC. The initial step in the synthetic sequence utilised the trimethyltin precursor **62** (Figure 19). This precursor was synthesised from *N*-Boc-protected 3-iodo-7-azaindole which was initially mixed with trimethyltin chloride at −78 °C, before the addition of *n*-butyllithium yielded the 7-azaindolestannane **62** in 55–60%. Once **62** was prepared, it was reacted with 2,3-dichloro-*N*-methylmaleimide **15**, in a palladium catalysed C-C cross-coupling reaction [17].

The *N*-Boc groups were cleaved by the subsequent addition of trifluoroacetic acid at room temperature. Once the *N*-methyl maleimide **63** was successfully furnished, it was progressed to the alkylation step with the bismesylate **64** in the presence of caesium carbonate to afford the *N*-methylmaleimide macrocycle **65**. Following this, base hydrolysis afforded the maleic anhydride intermediate, which was not isolated, but rather underwent ammonolysis to give the target bisazaindolylmaleimide (**66**).

Kozikowski et al. demonstrated the incorporation of benzofuran into the bisindolylmaleimide frame using substituted benzofuran precursors in the coupling step [34]. A diverse panel of benzofuranylindolylmaleimides **69** (Figure 20) was synthesised via a Perkin-type condensation of substituted benzofuran-3-yl acetamides **68** with indol-3-yl oxoacetate esters **67** in yields ranging from 20 to 72%, depending on the solubility of the precursors. The lead compounds, **70** and **71**, were prepared as part of this synthetic series [35].

Another modification of interest was to retain one indole component and to introduce naphthalene into the BIM framework. In 1999, a variety of naphthylindolylmaleimides (NIMs) were first synthesised and evaluated [36]. This involved a similar synthetic methodology to that utilised to generate arcyriarubin A (**BIM-IV**) (Figure 13) and enzastaurin (**5*7***) (Figure 17) with naphth-1-ylacetamide **72** employed in place of indole. This was condensed with indole oxoacetate ester **42**, successfully generating the NIM **73** in a high yield (Figure 21).

In 2017, van Eis et al. employed an analogous route to access a series of naphthyl-functionalised indolylmaleimides (Figure 22) [37]. The starting material, 7-methoxy-3,4-dihydronaphthalen-1(2*H*)-one **74**, was carried through a series of synthetic steps to form **75**. Condensation of **75** with the indole oxoacetate ester framework **76** generated the maleimide scaffold **77**. The best candidate for PKCα/β inhibition was the maleimide **78**, where both R^1^ and R^2^ were methyl, and this was generated in 11% overall yield following the 11-reaction-step sequence.

To summarise, many synthetic routes towards the bisindolylmaleimide pharmacophore have been reported and each presents advantages or synthetic challenges. Initially, maleimide substitution was explored with the Grignard approach, which required the capping of the maleimide nitrogen and thus was non-optimal for accessing the bioactive N-H maleimides (Figure 1). There were similar limitations associated with the palladium coupling route, and low yields were also obtained (Figure 7 and Figure 8), though more recent efforts involving maleimide substitution have overcome these (Figure 9). Alternately, combining two indole units through a Perkin-type condensation step to produce the maleimide has proved more robust towards N-H maleimides (Figure 10 and Figure 13). N-substitution on the indole precursors must be carefully selected in order to design the most efficient route to the desired substitution pattern. It is interesting to note that both methodologies have been used in the synthesis of clinically relevant BIMs and derivatives and signals that structural modification of the BIM pharmacophore is of eminent importance in the field. The synthetic methodology outlined proves that this pharmacophore is highly accessible, and that substituent choice can direct activity towards a biological target of interest.

## 3. Bioactivity of Bisindolylmaleimides and Derivatives

The bioactivity of bisindolylmaleimides and their derivatives began with the discovery in 1980 of Acyrirubin A (**BIM-IV**) and has led to new discoveries and clinical agents for the treatment of disease, in addition to their use in other fields of research, such as sensors. This section will focus on BIM compounds and specifically those with reported clinical effects.

### 3.1. Bisindolylmaleimides (BIMs) and BIM-Type Inhibitors

Arcyriarubin A (**BIM-IV**) is the simplest bisindolylmaleimide that belongs to the family of pigments, arcyriarubin A-C (Figure 2). It was isolated from the *Myxomycetes* slime moulds by Steglich et al. in 1980 [9]. **BIM-IV** was a potent sub-micromolar inhibitor of protein kinase C and exhibited micromolar inhibition against seven of the other PKC isoenzymes. Fabre et al. investigated the influence of the maleimide headgroup on **BIM-IV** against PKC and PKA by comparison with the succinimide **79** and the lactam derivative **80** [38]. Both **79** and **80** were determined to have low inhibitory activity compared to **BIM-IV** against both PKC and PKA (Figure 2).

In subsequent studies, lactam **81** exhibited more potent activity against PKC than against PKA in micromolar concentrations [19,39]. These studies reveal the critical nature of the maleimide ring to the kinase inhibition of arcyriarubin A (**BIM-IV**). Additional antimicrobial screening of **BIM-IV** revealed that unlike the indolocarbazoles staurosporine and K-252a, it inhibited sporulation and inhibited the growth of *Streptomyces chartreusis* and *Streptomyces griseus* [40].

Interest subsequently shifted towards *N*-alkylated indole subunits to further probe kinase inhibition. In 1990, Toullec et al. investigated the activity of the bisindolylmaleimide GF109203X (**BIM-I**) against PKC and five other protein kinases (Figure 3) [41].

GF109204X (**BIM-I**) was quickly recognized as an inhibitor of specific PKC isoforms at nanomolar concentrations (Figure 3). It was also discovered as a competitive inhibitor of ATP (*K_i_* = 14 nM) and it efficiently halted PKC-mediated phosphorylation and successfully inhibited collagen-triggered ATP secretion and collagen- and thrombin-induced platelet aggregation. As well as being used as a standard molecular tool to explore the role of PKC in disease, more recently GF109203X has been identified as a potent agonist of β-catenin accumulation in preosteoblast cells, promoting osteoblast differentiation and bone formation (through suppression of GSK-3β kinase) [42,43]. Another new application was identified in 2017 in the inhibition of exosome and microvesicle release to improve the efficiency of cancer treatment [44].

In 2005, Graff et al. reported the dialkylated bisindolylmaleimide enzastaurin (LY317615.HCl) (**57**) as an ATP-competitive inhibitor of PKC (Figure 4). It was found to exhibit potent activity against the PKC isoforms α, β, γ and ε, with some selectivity reported for PKC-β [45]. The interaction between PKC and the phosphatidylinositol 3-kinase (PI3K)/AKT pathway is consistent with the fact that **57** interferes with AKT pathway signalling and acts through mechanisms with both direct and indirect antitumour effects, such as the direct induction of apoptosis and suppression of tumour cell proliferation, or by indirectly halting tumour-induced angiogenesis [46]. In early clinical trials, enzastaurin (**57**) was reported as an oral serine/threonine kinase inhibitor which selectively targets PKC-β and the PI3/AKT signalling pathways. Enzastaurin suppressed angiogenesis with a reduction in the growth of human glioblastoma and colon carcinoma xenografts [45]. The drug was granted orphan drug status for the treatment of diffuse large B-cell lymphoma (DLBCL) in 2007; clinical trials were halted after limited efficacy as a monotherapy for cancer was reported [45,47].

Having completed phase III, no toxicity was reported when used for the treatment of solid tumours and hematologic malignancies involving >3000 patients, and thus enzastaurin has a well-mapped safety profile. This has led to a focus on drug repurposing and licensing for specific uses under various names (ENZA, DB102, AR101 and Kinenza). In 2020, enzastaurin was granted Fast Track designation by the FDA for the treatment of newly diagnosed glioblastoma multiforme with biomarker DGM1 (ENGINE clinical trial in combination with R-CHOP and the ENGAGE clinical trial with Temozolomide), and in 2022 it was granted orphan drug status in the US and safe-to-proceed through the PREVEnt clinical trial for patients with COL3A1 mutation with vascular Ehlers–Danlos Syndrome [48,49,50,51]. It is clear that Enzastaurin has a clinical role to play in future.

Expanding on the theme of *N*-alkylation, macrocyclic bisindolylmaleimides were investigated (Figure 5). Initially, the *N–N′* bridged alcohol **60** (LY326449) was prepared [32]. Although there was a noticeable improvement in activity against PKC-α and PKC-β_2_ and it showed >10,000-fold selectivity for PKC over PKA, it did not progress to the clinic [52]. Subsequent incorporation of dimethylamine led to the important discovery of ruboxistaurin (LY333531) (**61**). This compound was found to be a potent and selective nanomolar inhibitor of PKC-β_2_. Given these significant results, the activity of **61** was evaluated against other PKC isoforms, but potent inhibition was only reported for PKC-β_1_ (IC_50_ = 0.0047 μM) [53]. It was found to interact at the ATP binding site to disrupt the phosphotransferase activity of novel and conventional PKC isoforms [54].

Further studies with **61** revealed that PKC activation of the β isoenzyme, in particular, contributes to vascular complications, including hemodynamic changes associated with diabetes, and thus PKC-β inhibition may be a promising approach. The efficacy of ruboxistaurin was evaluated for an anticancer effect as a combination therapy, which led to the enhanced activity of paclitaxel but also carmustine and when co-administered with **61** in separate assays. It progressed to phase 1 oncology trials and phase 2/3 for diabetic neuropathy, with no further progress reported after 2006, and a withdrawal report was issued in 2007 [55,56,57]. However, as with enzastaurin, artificial intelligence has identified new applications and ruboxistaurin is currently under evaluation by Dermbiont (as DBI-102, a topical application for hyperpigmentation) and Recursion (as REC-3599 for the treatment of Tay–Sachs disease or GM2 gangliosidosis) [58].

Kuo et al. investigated the inhibition of PKC-γ in order to access new therapies for chronic pain [33]. Macrocyclic bisindolylmaleimides that mimic ruboxistaurin (**61**) with the incorporation of the 7- azaindole moiety was reported (Figure 6). Initially, the bisindole system and mono-7-azaindole derivatives were prepared and evaluated. Following this, bis-7-azaindolylmaleimide was generated with ether chains of varying lengths (n).

Upon evaluation of their activity, **61** was still more potent for PKC-γ (IC_50_ = 0.3 μM) than any of the macrocyclic candidates in this synthetic library (Figure 6). The most interesting finding was the potency against GSK-3β at sub-micromolar concentrations. By introducing the bis-7-azaindole core (**86** and **66**), the activity shifted towards exclusive GSK-3β inhibition and lead compound **66** showed nanomolar inhibition of GSK-3β (IC_50_ = 0.034 μM). Although a slight extension of the macrocyclic chain retained activity, longer chains, e.g., **88** (where n = 4), significantly diminished activity against GSK-3β.

Following further evaluation of **66**, no competing activity for other protein kinases was identified, with the PKC-β_2_ isoenzyme as the only exception (Figure 7). Lead compounds **66** and **87** were further screened in a broad 51-protein kinase assay to assess their degree of selectivity. They both exhibited minimal or no inhibition of other kinases in the screen, effectively inhibiting GSK-3β activity by 100% at 10 μM and identified as potential specific GSK-3β inhibitors. A glycogen synthase (GS) assay was also conducted to compare the activity of known GSK-3β inhibitor LiCl with the two lead candidates. Both aza-compounds demonstrated greater potency than LiCl (EC_50_ > 3000 μM), where values of 0.06 μM and 0.39 μM were measured for **66** and **87**, respectively. The selectivity and potency demonstrated by these two aza-BIMs has led to a better understanding of GSK-3β in signalling pathways associated with GSK-3β-induced disorders and to the further development of BIM-like molecules.

### 3.2. Selected Modified Indolylmaleimides: Benzofuranylindolylmaleimides (BfIMs)

The initial incorporation of other aryl units, such as 7-azaindole, into the BIM frame proved effective in enhancing inhibitory selectivity against kinases. Benzofuran was also an attractive heterocyclic component, which was initially employed by Davis et al. to prepare the maleimide **89** [59]. Similar to the BIMs discussed earlier in the chapter, capping the indole nitrogen with an *N*-methyl group increased potency for this novel class of benzofuranylindolylmaleimides (BfIMs). Significant PKC inhibition (IC_50_ = 200 nM) was reported for this *N*-methyl compound (Figure 8).

Over ten years later, Kozikowski et al. investigated a panel of 21 BfIM compounds and assessed their ability to inhibit GSK-3β [34,35]. From a structural perspective, 5-, 6- and 7-substitution (R^1^) on the indole unit ranged from halogens to bulky groups in order to gauge tolerance, the majority of candidates contained an *N*-methyl group on the indole nitrogen (R^2^) and substituents were also installed on the benzofuran (R^3^) (Figure 8). Overall, there is broad acceptance of substituents, aside from large steric bulk, and low nanomolar inhibition is seen with small halogen and H-bonding groups on the indole. On the benzofuran unit, a 6-hydoxymethyl group elicited inhibition at sub-nanomolar concentrations. Two compounds in particular were explored further, methylenedioxy **70** (710 nM) and di-halogenated indole BfIM **71**, which demonstrated potent nanomolar inhibition of GSK-3β (3.5 nM). In order to visualize binding in the active site, **71** was co-crystallized with the GSK-3β kinase, as shown in Figure 9. The conformation of compound **71** in the active site is not planar, but rather, two heteroaromatic units that are orientated parallel to one another. The maleimide headgroup is confirmed to H-bond to the protein backbone of Asp133 and Val135 and given the orientation of the aromatic rings, there appears to be sufficient space for diverse substituents, as seen in the GSK-3β kinase inhibition measurements.

Further analysis of compound **71**, as well as the methylenedioxy compound **70**, identified the relationship between GSK-3β kinase inhibition and pancreatic cancer cell lines.

Although the methylenedioxy compound **70** was not the most potent compound, it was chosen as the clinical candidate (9-ING-41) for its broad spectrum, pre-clinical antitumour activity [60]. Orphan drug status was granted for the BfIM **70** by the FDA for the treatment of neuroblastoma, as it is a potent growth suppressor of neuroblastoma cells through GSK-3β inhibition. In 2018, the FDA approved this ATP-competitive inhibitor for phase I/II clinical trials for patients with advanced cancer (clinical trial no. NCT03678883) [61]. This therapeutic candidate exhibited significant activity and low toxicity in both phase I and II, which have been successfully completed.

Following this, Jeffers et al. investigated **70** as a potential treatment for bleomycin-mediated pulmonary fibrosis (PF) as myofibroblast differentiation and pulmonary fibrosis are induced by the GSK-3β signalling pathway, ex vivo and in vivo, respectively [62]. It was discovered that the GSK-3β inhibitor significantly improved lung function in mice treated with TGF-β adenovirus and also bleomycin-induced PF mice models. In 2020, Anraku et al. reported a broader antiproliferative scope of 9-ING-41 (**70**) against renal cancer cell lines [63]. It effectively induced cell cycle arrest and apoptosis as a single agent, but also proved effective in combination with standard therapies to improve antitumour effects.

### 3.3. Selected Modified Indolylmaleimides: Napthylindolylmaleimides (NIMs)

As seen earlier, the derivatisation of the BIM frame dates back to 1992 when Davis et al. replaced one of the indole units with aryl components [59]. A wide panel of compounds were generated with aryl systems, including substituted phenyls, and thienyl and pyrrolyl groups. In 2006, Peifer et al. investigated arylindolyl-2,3-maleimides and evaluated their antiangiogenic activity in an in vivo assay with chick embryos. As part of this synthetic panel, naphthyl-containing maleimides were included to assess their inhibitory potential against protein kinases (Figure 10) [64]. All compounds were screened against twelve kinases with close attention to CDKs and PKC isoenzymes. Naphthylindolylmaleimide **73** was identified as a potent inhibitor of PKC-β_1_ (IC_50_ = 2 nM).

The NIMs were developed further in 2017 by van Eis et al. by modifying the naphthalene component at the 2-, 6- and 7-positions to evaluate biological activity in T and B cell proliferation assays [37]. These were tested against the conventional and novel PKC isotypes, and initial assessment of the 2,6-substituted naphthalenes yielded potent PKC inhibition but little isoform selectivity. Movement of the dimethylamino methylene chain to the C7 position of the naphthalene ring (**78**) yielded remarkable PKC-α (0.5 nM) and PKC-β (0.7 nM) selectivity over PKC-δ, ε, η and θ (>182 nM). Modification of the amine was detrimental to the activity and it was found that compound **78** provided an optimal balance between potency and selectivity when screened against a broad panel of 136 kinases.

### 3.4. Recent Applications of Bisindolylmaleimides and Derivatives

In recent years, non-PKC targeted effects for BIMs have come to the fore with a number of examples of repurposing existing BIMs and the development of new BIMs to align with new targets. **BIM-IX** was identified by Zhang et al. to affect drug-resistant chronic myeloid leukaemia (CML) by inhibiting DNA topoisomerase and inducing cell cycle arrest and cell death [65]. It appears that **BIM-IX** is more effective than enzastaurin and other BIMs against BCR-ABL positive and T315I mutated cells and maintains its effect on in vivo cancer models through inhibition of topo IIa and B-Raf.

On a similar note, both Li and Winfield et al. identified new BIM compounds with diverging activity. Li identified that the active BIM **90** bound to the SH2 domain of STAT3, and that substitution of the maleimide NH with hydroxymethyl eliminated this interaction, whereas Winfield identified that kinase inhibition could be modified by its substitution to N-OH **91**. In both papers, novel *N*-alkylated BIMs were identified to interact with STAT3 and kinases through screening, and it is remarkable that the most active compounds contain alkyl nitrile substituents on the indole nitrogens (**90** and **91**, Figure 11) [28,29]. Again, moving away from PKC, Mayati et al. identified inhibition of the organic cation transporter 1 (OCT1) by **BIM-IX** (Ro 31-8220). This has important considerations for the activity of BIMs in drug-resistant cells and should especially be considered for other BIMs in relation to the likelihood of cellular off-target effects [66].

Another example of new targets is the identification of BIMs as inhibitors of calmodulin protein [67]. In this study, standard BIM compound targets are summarised as in Table 1, and it is noted that most bioassays are conducted where the reported effects could arise from interaction with more than one molecular target. In testing a series of BIMs binding to calmodulin, **BIM II**, **IV**, **VII**, **X** and **XI** were identified with nanomolar affinity and have the potential as the starting point of new calmodulin inhibitors.

As a part of the enormous research effort to combat the COVID-19 epidemic, BIMs have also been identified to possess protective activity against SARS-CoV-2 infection. Gupta et al. identified 290 potential inhibitors from a high throughput virtual screen of 5903 molecules against seven essential coronavirus enzymes [76]. **BIM IX** proved the most successful and the target enzyme responsible for this effect was confirmed as the protease 3CL^pro^. While the compound was not as effective as ivermectin, this could prove an exciting lead compound for future studies. Following this, Huang et al. screened four PKC inhibitors: **BIM I**, a derivative of BIM I (Go 6983, Figure 12), enzastaurin and sotrastaurin for antiviral activity in a SARS-CoV-2 replicon system, and identified that all compounds reduced viral replication and all but sotrastaurin had antiviral activity against wild-type SARS-CoV-2 [77].

Finally, although the majority of BIM compounds maintain the integrity of the maleimide imide as a key H-bonding component, there are some recent reports of *N*-substitution to achieve non-kinase effects. In 2017, Sun et al. synthesised a number of new BIM compounds of which BMA-155 and its hydrochloride salt (Figure 12) were identified as potent anticancer compounds operating through the NF-kB p65 pathway and effecting apoptosis both in vitro and in vivo [78]. BMA-155 is modified at the maleimide nitrogen by ethylamine substitution so this template opens up new structural avenues in BIM bioactivity. More recently, in 2022, Kumar et al. generated a series of aminoalkyl-substituted BIMs which incorporate substitution on both indole nitrogens and the maleimide nitrogen [79]. These compounds were reported to preferentially stabilise the G-quadruplexes of *c-MYC* and *c-KIT* and provide a new starting point for the development of BIM ligands capable of regulating gene expression.

Finally, aside from clinical and pharmacological endpoints, there exist other BIM targets which present opportunities in fields such as sensors [80].

### 3.5. X-ray Crystal Structures of Bisindolylmaleimides and Kinases

As seen in Section 3.2, the kinase-targeted effects of BIMs can be rationalized through crystal structure formation as ligands. To date, X-ray crystal structures have been solved for BIM compounds in a number of kinases, as detailed in Table 2. It is clear that BIM compounds are capable of forming crystal structures across the kinase families and, interestingly, the two most often quoted targets of BIMs PKC-β and GSK-3β are in the minority of solved structures. This, however, does not prevent the use of modelling to plot interactions, and indeed high throughput in silico screening has been used extensively in the last 10 years to develop the scope of targets and to rationalize the effects of these potent compounds.

## 4. Conclusions

To summarize, bisindolylmaleimides and their derivatives can be synthesized in good yields and with enormous potential for diversity. This diversity of synthesis in turn allows for a high level of structural tunability and hence offers the potential for divergent bioactivity within the series. Looking at the starting BIM standards (**BIM I**-**XI**), it is clear that PKC kinase and protein kinases in general are noted targets, and so any new BIM derivative should be screened for this activity. It is also evident that multiple targets can be effectively inhibited by compounds with the bisindolylmaleimide pharmacophore and indeed selectivity within the kinases can be achieved, which suggests there is significant value in further development.

Since the early 1990s, both acyclic and macrocyclic BIM agents have excelled during in vitro studies and progressed to in vivo testing and human clinical trials. The introduction of different aryl units to the BIM pharmacophore proved to be a successful strategy to enhance target selectivity. The BIM class offers unique opportunities for drug repurposing due to its known safety profile and indeed many BIMs have completed clinical trials, which is evidence of their real-world impact. Beyond the BIM structures we have investigated, there are many other aryl maleimides [95] and bisindole derivatives [96,97,98] which have notable bioactivity, and hence this expanding field is sure to maintain real-world relevance in the future.

The prediction of the biological targets for these molecules is challenging due to incomplete literature with respect to bioactivity screening as, in the main, compounds are screened against a limited number of targets. However, the impact of AI on the field is beginning to show promise with a number of new clinical trials stemming from this in the past few years, and this approach has exponential potential. This information and predictive power will greatly assist the design and discovery of new, highly potent and specific drug candidates. Overall, it is clear that the BIM class of compounds continues to be clinically relevant and its future development is keenly anticipated by the research community.

## Data Availability

All data are freely available through the references quoted within.

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
