# Peer review of "Bisindolyl Maleimides and Indolylmaleimide Derivatives—A Review of Their Synthesis and Bioactivity"

_pharmaceuticals, 2023, doi:10.3390/ph16091191_

Round 1

Reviewer 1 Report

This manuscript reviewed the synthetic approaches to bisindolylmaleimides (BIMs) and BIM derivatives, as well as some of their associated bioactivities. It is of great interest to researchers in related fields. However, this manuscript should be refined further.

1.      BIMs were reported with a variety of biological activities, such as inhibition of GSK3, antagonist of 5-HT3 receptor, and interaction with ABCG2, besides protein kinase inhibition. The authors should supplement the details and references in the introduction section.

2.      This reviewer suggests to combine and adjust partial content of the paper. For example, it is better that the description of the synthesis of compound 57 or 61 could be followed by the part of the bioactivities of these compounds for the sake of readability.

3.      There are some mistakes needed to correct.

“N-benzyl-3,4-dibromomaleimide” should be “N-benzyl-2,3-dibromomaleimide”;

“Scheme 2.11: Preparation of maleimide 37 from the maleic anhydride” should be “Scheme 2.11: Preparation of maleimide 36 from the maleic anhydride”

Author Response

Reviewer 1 Report Form

We thank the reviewer for their time and expertise in reviewing our submission and appreciate the value of each comment. 

Comments and Suggestions for Authors

This manuscript reviewed the synthetic approaches to bisindolylmaleimides (BIMs) and BIM derivatives, as well as some of their associated bioactivities. It is of great interest to researchers in related fields. However, this manuscript should be refined further.

  1. BIMs were reported with a variety of biological activities, such as inhibition of GSK3, antagonist of 5-HT3 receptor, and interaction with ABCG2, besides protein kinase inhibition. The authors should supplement the details and references in the introduction section.

We thank the reviewer for the suggestion but the scope of the review is set out in the introduction along with references to any previous reviews related to the field. All subsequent sections detail the original research articles from which the field is comprehensively reviewed including the areas beyond kinase inhibition. We have added a sentence to capture this more clearly: P2 Line46  “In Section 2 we will summarise the methods used to make diverse BIMs, and Section 3 the recorded biological activity of diverse BIMs uncovering kinase and non-kinase effects and with a focus on their solved crystal structures and clinical applications.” If there are other specific review articles related to the field of bisindolylmaleimides which have been omitted we can include in the introduction.

  1. This reviewer suggests to combine and adjust partial content of the paper. For example, it is better that the description of the synthesis of compound 57 or 61 could be followed by the part of the bioactivities of these compounds for the sake of readability.

This approach has merit but would require a total reworking of the manuscript, the format of which has been set out clearly in the introduction.

  1. There are some mistakes needed to correct.

“N-benzyl-3,4-dibromomaleimide” should be “N-benzyl-2,3-dibromomaleimide”

Thanks for spotting this - this has been amended as directed

“Scheme 2.11: Preparation of maleimide 37 from the maleic anhydride” should be “Scheme 2.11: Preparation of maleimide 36 from the maleic anhydride”

Thanks for spotting this - this has been amended as directed

Reviewer 2 Report

this paper is good quality work and may be accepted after minor revision 

1. work needs to add more references like the below examples 

a)https://doi.org/10.1007/s11030-020-10084-4

b)https://doi.org/10.1007/s00044-019-02431-4

2. add more information in the abstract and conclusion.

3. a lot study has been conducted on the biological properties of bisindole which details are missing here.

4. check the references according to the journal style 

Author Response

Reviewer 2 Report Form

We thank the reviewer for their time and expertise in reviewing our submission and appreciate the value of each comment. 

Comments and Suggestions for Authors

this paper is good quality work and may be accepted after minor revision 

  1. work needs to add more references like the below examples 

a)https://doi.org/10.1007/s11030-020-10084-4

b)https://doi.org/10.1007/s00044-019-02431-4

We thank the reviewer for these suggestions but the reference examples provided are not within the scope of the review as set out (bisindole methyls as opposed to maleimides). To incorporate these would require a whole new review and can be considered in future.

  1. add more information in the abstract and conclusion.

More details have been added to the conclusion

  1. a lot study has been conducted on the biological properties of bisindole which details are missing here.

Again, the scope of this review is a focus on bisindolylmaleimides and other bisindole compounds have been omitted. These could form part of a separate future review.

  1. check the references according to the journal style 

Checked

Reviewer 3 Report

The review manuscript entitled “Bisindolyl maleimides and indolylmaleimide derivatives – a review of their synthesis and bioactivity” is nicely described about the various synthetic routs and broad spectrum of pharmacological importance of BIMs (Bisindolyl maleimides).  

The authors can correct these minor mistakes.

1. In figure 1.1, there is no structural diversity in first two rows. Almost very similar BIMs are presented. It would be more informative to replace some of BIMs like BIM-III with GSK-3β inhibitor (66) (Scheme 2.19); BIM-XI with BfIMs (70 or 71) (Scheme 2.20).

2. Though, the structures of Ruboxistaurin in figure 1.1 (LY 333531) and Scheme 2.18 (compound 61) are same, but drawn differently. It would be more appropriate to draw the structures in single format.

3. I think the word “Unfortunately” is not suitable to use in line 119, page 5 and can be replaced with “But”. It diminish the efforts of the authors Foul et al. (1995). Grignard reaction is one of the easiest way to prepare in bulk scale.

4. The sub headings 2.1 to 2.4; 3.1 to 3.5 are not clearly visible. It would be better to increase the font size or put in bold format.

5. In the Bioactivity section 3, the authors can incorporate more images like in Figure 3.8, to increase the reader’s attention.

6. In Reference no. 5, please remove brackets for volume number 82.

7. In Reference no. 6, the authors simply gave the article published year and doi link. It should be Nat. Prod. Rep., 2021,38, 1794-1820.

8. In reference no. 13, please remove “,” after the volume no. 34.

In references, the authors can give issue numbers to every reference, beside the volume numbers or can give only volume numbers. It would be better to follow one format. Also remove the doi numbers, where there is a clear publication details.

Other than these corrections, the review manuscript looks very good and well explained. I strongly recommend to accept this review after these minor corrections.

Author Response

Reviewer 3 Report Form

We thank the reviewer for their time and expertise in reviewing our submission and appreciate the value of each comment. 

Comments and Suggestions for Authors

The review manuscript entitled “Bisindolyl maleimides and indolylmaleimide derivatives – a review of their synthesis and bioactivity” is nicely described about the various synthetic routs and broad spectrum of pharmacological importance of BIMs (Bisindolyl maleimides).  

The authors can correct these minor mistakes.

  1. In figure 1.1, there is no structural diversity in first two rows. Almost very similar BIMs are presented. It would be more informative to replace some of BIMs like BIM-III with GSK-3β inhibitor (66) (Scheme 2.19); BIM-XI with BfIMs (70or 71) (Scheme 2.20).

Thank you for your comments – the purpose of Figure 1.1 is to catalogue the named BIM compounds which are commonly used as standards in literature and are available commonly in chemical catalogues. We appreciate the need to include diversity and have modified the graphical abstract appropriately.

  1. Though, the structures of Ruboxistaurinin figure 1.1(LY 333531) and Scheme 2.18 (compound 61) are same, but drawn differently. It would be more appropriate to draw the structures in single format.

Thank you for spotting this – we have amended all ruboxistaurin (and analogous structures) alike

  1. I think the word “Unfortunately” is not suitable to use in line 119, page 5 and can be replaced with “But”. It diminish the efforts of the authors Foul et al. (1995). Grignard reaction is one of the easiest way to prepare in bulk scale.

This has been removed

  1. The sub headings 2.1 to2.4; 3.1 to3.5 are not clearly visible. It would be better to increase the font size or put in bold format.

The format of the journal has been adhered to for these headings

  1. In the Bioactivity section 3, the authors can incorporate more images like in Figure 3.8, to increase the reader’s attention.

This is a good suggestion – in Table 2 we have provided every PDB code for an X-ray crystal structure incorporating a BIM where the interactions can be seen in 3D.

  1. In Reference no. 5, please remove brackets for volume number 82.

Thank you for spotting this – this is amended

  1. In Reference no. 6, the authors simply gave the article published year and doi link. It should be Nat. Prod. Rep., 2021,38, 1794-1820.

Again, thank you for spotting this - it has been amended

  1. In reference no. 13, please remove “,” after the volume no. 34.

These do not appear on my version of the manuscript but we will check again the final version in case it is a formatting issue

In references, the authors can give issue numbers to every reference, beside the volume numbers or can give only volume numbers. It would be better to follow one format. Also remove the doi numbers, where there is a clear publication details.

All references will be formatted as per the journal instructions

Other than these corrections, the review manuscript looks very good and well explained. I strongly recommend to accept this review after these minor corrections.